# Advances in the Regulation of Mammalian Follicle-Stimulating Hormone Secretion

**DOI:** 10.3390/ani11041134

**Published:** 2021-04-15

**Authors:** Hao-Qi Wang, Wei-Di Zhang, Bao Yuan, Jia-Bao Zhang

**Affiliations:** Department of Laboratory Animals, Jilin Provincial Key Laboratory of Animal Model, Jilin University, Changchun 130062, China; hqwang19@mails.jlu.edu.cn (H.-Q.W.); zhangwd19@mails.jlu.edu.cn (W.-D.Z.)

**Keywords:** pituitary, follicle-stimulating hormone, gonadotropin-releasing hormone, signal transduction, animal reproduction

## Abstract

**Simple Summary:**

The reproduction of mammals is regulated by the hypothalamic-pituitary-gonadal axis. Follicle stimulating hormone, as one of the gonadotropins secreted by the pituitary gland, plays an immeasurable role. This article mainly reviews the molecular basis and classical signaling pathways that regulate the synthesis and secretion of follicle stimulating hormone, and summarizes its internal molecular mechanism, which provides a certain theoretical basis for the research of mammalian reproduction regulation and the application of follicle stimulating hormone in production practice.

**Abstract:**

Mammalian reproduction is mainly driven and regulated by the hypothalamic-pituitary-gonadal (HPG) axis. Follicle-stimulating hormone (FSH), which is synthesized and secreted by the anterior pituitary gland, is a key regulator that ultimately affects animal fertility. As a dimeric glycoprotein hormone, the biological specificity of FSH is mainly determined by the β subunit. As research techniques are being continuously innovated, studies are exploring the underlying molecular mechanism regulating the secretion of mammalian FSH. This article will review the current knowledge on the molecular mechanisms and signaling pathways systematically regulating FSH synthesis and will present the latest hypothesis about the nuclear cross-talk among the various endocrine-induced pathways for transcriptional regulation of the FSH β subunit. This article will provide novel ideas and potential targets for the improved use of FSH in livestock breeding and therapeutic development.

## 1. Introduction

Follicle-stimulating hormone (FSH) is a glycoprotein hormone synthesized and secreted by the pituitary gland. The pituitary gland, as one of the endocrine organs of the animal, plays a crucial and pivotal role in different physiological processes of mammals due to the secretion of various hormones. In addition to FSH, the other hormones secreted by the pituitary gland are growth hormone (GH), prolactin, adrenocorticotropic hormone, melanocyte-stimulating hormone, thyroid-stimulating hormone (TSH) and luteinizing hormone (LH) [1]. Among these hormones, FSH, as one of the important gonadotropins involved in mammalian reproductive development, is secreted into the blood after synthesis by the gonadotroph cells (a type of basophilic cell) in the anterior pituitary gland (adenohypophysis). Then, it acts on the corresponding target organs of the mammals, namely the testes and ovaries, to exert its biological functions through the peripheral blood circulation [2]. Since FSH is a key regulator in the hypothalamic-pituitary-gonadal (HPG) axis, it plays an indispensable role in mammalian reproductive activities.

Some reproductive disorders are associated with the disruption of FSH secretion, and/or its signaling pathways. For instance, it has been observed that the concentration of FSH was lower in polycystic ovarian syndrome (PCOS) than in the controls [3]. In women with PCOS, it can promote follicular development by injecting an appropriate amount of exogenous FSH to supplement the low concentration of FSH caused by insufficient endogenous secretion [4,5]. The lack of FSH and FSHR may also cause difficulty in spermatogenesis and infertility in men [6]. Some preliminary data suggest a beneficial effect on live birth and pregnancy of gonadotrophin treatment for men with idiopathic male factor subfertility [7,8].

In view of the non-negligible role of FSH in mammalian reproductive development, it is very meaningful to learn about how to regulate the synthesis and secretion of FSH. In this review, we summarize the classical molecular characteristics and signaling pathways involved in the regulation of mammalian FSH secretion.

## 2. Function and Structure of FSH

FSH and luteinizing hormone (LH) synergistically regulate animal reproduction through specific G protein-coupled receptors (GPCRs) under physiological conditions, and they can also regulate steroid hormone production, cell metabolism and growth and other physiological activities, thereby exerting specific biological effects on the hypothalamus, pituitary, ovary, testis and other target tissues [9,10,11,12]. The formation and maturation of ovarian follicles, the proliferation of follicular granulosa cells, the synthesis of sertoli cells and leydig cells, and the development of seminiferous epithelium all require the cooperation of gonadotropins. FSH plays different functions in female animals and male animals.

In female animals, FSH stimulates the growth and development of follicles, and increases the oxygen uptake of parietal granulosa cells to promote related protein synthesis [13]. Especially in the late stage of follicle formation, FSH induces granulosa cells to express a large number of luteinizing hormone receptors and their own proliferation and induces an increase in the expression of epidermal growth factor receptor (EGFR) to promote the occurrence of ovulation under the synergistic effect of LH [14,15]. In addition, FSH treatment up-regulated the synthesis of related hormones, including progesterone [16]. The clinical manifestation that FSH-deficient women become infertile due to blocked follicle production also implies the essential role of FSH [17]. FSH can also promote differentiation of the follicular inner membrane cells, thereby promoting the proliferation of granulosa cells and the secretion of follicular fluid [18].

In male animals, FSH promotes seminiferous epithelial development and spermatogenesis [19]. Congenital FSH deficiency caused by the *FSHB* mutation could directly lead to abnormal sperm and even infertility [6]. With the emergence of recombinant FSH, more and more FSH preparations or biosimilar drugs are used in the treatment of male infertility, while the use of different FSH preparations achieved similar results in stimulating spermatogenesis in males and eventually inducing physiological pregnancy [20]. In addition, FSH has a direct effect on germ cells, such as supporting cells and spermatogonial stem cells in the testis [21,22]. The development of testes and the synthesis of testosterone are also inseparable from the participation of FSH [23].

Due to in-depth studies of FSH functions, new FSH functions are gradually being recognized in addition to the traditional physiological functions. For example, FSH may regulate the endocrine function of the rat pancreas via the FSHR [24]. FSH has also been gradually confirmed to play a potential role in bone [25], fat [26], prostate tumors [27] and other tissues. However, the detailed mechanism still needs further analysis.

FSH is a heterodimeric glycoprotein consisting of two noncovalently bound and dissociable subunits, α and β [28], and its molecular structure is similar to that of LH. For this type of glycoprotein hormone, the α-subunit is common, but the β-subunit has hormone specificity. Therefore, the β-subunit determines the biological specificity of gonadotropins, and the transcriptional differences of genes encoding the β-subunit will directly affect the synthesis and secretion of hormones [29]. It is well known that *FSHB* is highly expressed in the pituitary gland, but an increasing number of studies have found that it is also expressed in many tissues other than the pituitary gland. In 2010, Chu et al. [24] confirmed that *Fshb* and its receptor FSHR can be expressed in the rat pancreas. In addition, FSHR has also been identified in the female reproductive tract and developing placenta, and a low-level expression of *FSHB* has been detected in the following various nonovarian tissues: gravida, maternal decidua, placenta and myometrium [30]. This also means that FSH has many unknown functions to be explored, not just limited to the HPG axis.

## 3. The Molecular Basis of FSH Synthesis Regulation

### 3.1. Gonadotropin-Releasing Hormone

Gonadotropin-releasing hormone (GnRH) is the main regulator of FSH secretion as a decapeptide released by the hypothalamus [31]. GnRH is synthesized in hypothalamic neurons and secreted into the hypophyseal portal circulation. It mainly acts on anterior pituitary gonadotropin cells and binds to GPCRs on the cell surface, named gonadotropin-releasing hormone receptors (GnRHRs). Then, downstream signaling will be initiated, thereby inducing FSH synthesis [32]. GnRH is released in a pulsed manner under physiological conditions such as the regulation by kisspeptin, neurokinin B (NKB), dynorphin (Dyn), γ—aminobutyric acid (GABA) and glutamate [33,34]. Moreover, several sex hormones, such as estradiol (E2), also have a certain regulatory effect on the release of pulsed GnRH [35].

The expression of *FSHB* changes in response to the GnRH pulse frequency and amplitude changes [36]. This means that different GnRH pulse frequencies have different effects on the synthesis and release of FSH [37,38]. It is generally believed that GnRH pulse stimulation at a low frequency (maximum at an interval of every 120 min) preferentially promotes the transcription of the *FSHB* gene and then promotes the secretion of FSH; however, high stimulation of the GnRH pulse frequency (maximum at an interval of every 30 min) makes the transcription level of *FSHB* only slightly increase in the beginning. As the number of high-frequency stimulations increases, it gradually tends to have no significant changes and even appears to be weakly suppressed [39]. The reason for this phenomenon may be the heterogeneous effect of GnRH caused by a long-term or large-dose applications of GnRH or its highly-active analogs, such as Gonadorelin, Triptorelin and Leuprorelin. The above effects of GnRH can be blocked by GnRH antagonists, like Cetrorelix or Ganirelix, which can be used clinically to prevent premature ovulation [40]. GnRH pulse disorder can cause many diseases like hypothalamic amenorrhea and idiopathic hypogonadotropic hypogonadism (IHH) [41,42]. It is reported that GnRH pulse treatment successfully induces testicular growth and fertility in prepubertal testes of congenital hypogonadotropic hypogonadism (CHH) men and also has a significant effect on treating female IHH [42,43].

### 3.2. Kisspeptin

Kisspeptin is a peptide hormone encoded by the *KISS1* gene, which was first discovered in 2001 [44]. Kisspeptin and its G protein-coupled receptor KISS1R play key roles in mammalian reproduction due to control of the HPG axis [45]. The combination of kisspeptin/KISS1R and G protein subunit, Gq/11α activates phospholipase C (PLC), leading to the hydrolysis of phosphatidylinositol diphosphate and the formation of diacylglycerol (DAG) and inositol triphosphate (IP_3_), which in turn activates the downstream pathways, including the MAPK signaling pathway and cAMP signaling pathway [46,47]. The depolarization of kisspeptin neurons can also lead to the depolarization of GnRH neurons, and subsequently regulate the release of LH and FSH. In addition, NKB and Dyn, as the co-transmitters of kisspeptin signaling, cooperate with kisspeptin to regulate the release of GnRH and the synthesis and secretion of FSH [48]. At present, more research focuses on the following functions of kisspeptin: regulation of the surge and pulsatile center of GnRH in the hypothalamus [49], participation in feedback regulation of sex hormones [50,51,52,53], occurrence of puberty [54,55], control of reproductive behavior and ability and potential effects on reproductive diseases [56,57,58,59].

However, it is still unclear whether kisspeptin can directly regulate the related functions of the pituitary gland. It has been confirmed that kisspeptin can effectively induce the secretion of male LH, but the response of FSH to kisspeptin stimulation is much weaker [60,61]. This may be caused by differences in the secretion patterns of gonadotropins or differences in the response of different gonadotropin cells to kisspeptin. Therefore, a lot of research is still needed to explore the molecular mechanism of Kisspeptin, directly regulating gonadotropin. At present, the regulation of kisspeptin on FSH is still in the stage of indirect regulation through GnRH.

### 3.3. Activin and Inhibin

The transforming growth factor-β (TGF-β) family, a type of polypeptide growth factor widely present in various tissues ranging from Drosophila to humans, actively participates in the body’s regulation of FSH secretion. Activin and inhibin are two species of the TGF-β family that have received wide attention [62]. Activin plays an important role in inducing the expression of *FSHB*, increasing the release of FSH in the pituitary gland and regulating the concentration of FSH [63], which is named because of its ability to stimulate the pituitary gland to synthesize and secrete FSH [64]. A number of studies have confirmed that activin can activate the phosphorylation of drosophila mothers against decapentaplegic protein 2 (SMAD2) and drosophila mothers against decapentaplegic protein 2 (SMAD3) to mediate the transcription of *FSHB* [65,66]. Activin also participates in regulating the transcription of *FSHB* and the secretion of FSH through synergistic effects with GnRH [67]. Additionally, activin has a certain prolongation effect on the half-life of *FSHB* mRNA [68].

The role of inhibin, which is also a member of the TGF-β family, has been widely reported as a negative endocrine regulator of FSH in the HPG axis [69]. It was established that inhibin could selectively inhibit the secretion of FSH from pituitary gonadotropin cells, but there is no work on the secretion of LH between the 1930s and 1980s [70]. Subsequent studies have shown that inhibin plays a role in regulating the quantity of FSH that reach the follicles by inhibiting FSH-induced FSHR promoter activity and mRNA expression in female animals. In male animals, inhibin secreted by Sertoli cells is closely related to the sperm count [71], sperm concentration [72] and testicular volume [71]. Therefore, it is also considered to be the primary negative regulator of FSH in human males [73].

In addition to participating in the regulation of FSH, activin and inhibin also involve female reproductive diseases, such as being used as diagnostic biomarkers for granulosa cell tumors of the ovary and ovarian cancer [74,75].

### 3.4. Steroid Hormones

Steroid hormones have a certain regulatory effect on the secretion of FSH, which is referred to as feedback regulation due to the existence of the HPG axis. The regulatory mechanism of steroid hormones on FSH usually works together with multiple pathways mediated by GnRH, activin, inhibin and others [29]. Szabo et al. [76] found that the secretion of FSH mediated by activin was estrogen-dependent in 1998. Especially for female animals, different levels of steroid hormones, such as estradiol and progesterone in the body, have different effects on FSH secretion based on follicular development at different stages [77]. Many studies have confirmed that steroid hormones could act on the hypothalamus to indirectly regulate GnRH secretion or directly act on the pituitary and then affect *FSHB* expression, through positive feedback (high-estrogen-levels act on the cycle center of the hypothalamus to promote the secretion of GnRH and gonadotropin) or negative feedback (low-estrogen-levels act on the continuous center of the hypothalamus to continuously control the basic secretion of GnRH) [29].

However, it is controversial how estrogen specifically regulates GnRH secretion. As early as 2001, Herbison et al. [78] provided evidence for the hypothesis that GnRH neurons express estrogen receptors, which confirmed that GnRH neurons of rodents express estrogen receptor β (*ERβ*) mRNA during the entire developmental process. However, there are still studies indicating that GnRH neurons lack estrogen receptors, so the effect of estrogen on GnRH neurons is indirect [79]. At present, it is generally believed that estrogen regulates the function of GnRH neurons by acting on other neurons to release neurotransmitters. Furthermore, estrogen has been proven to have the potential function of changing the intracellular signal transduction mechanism. It is capable of affecting the phosphorylation of cAMP response element binding (CREB) and the phosphorylated CREB transfers to the nucleus where it dimerizes with cAMP responsive elements (CREs) or different leucine zipper partners in GnRH neurons to further regulate the function of GnRH neurons [80].

Steroid hormones can also regulate the synthesis and secretion of the hormone gonadotropin through other means. Estradiol and progesterone regulate the synthesis and secretion of LH and FSH, not only by regulating the release of GnRH, but also by enhancing the basic activity of the promoters of *Lhb* and *Fshb* in LβT2 cells [81]. Testosterone has also been confirmed to stimulate the release of FSH and maintain the intracellular level of FSH by activating the androgen receptor (directly or as dihydrotestosterone) to convert itself into estradiol or by directly activating the estrogen receptor [82]. In addition, glucocorticoids secreted by the adrenal glands can increase the expression of *FSHB* to selectively regulate the secretion of FSH [83,84].

### 3.5. Pituitary Adenylate Cyclase Activating Polypeptide

Pituitary adenylate cyclase activating polypeptide (PACAP) is named for its high activation of adenylate cyclase (AC) in rat pituitary cells and was originally thought to be a kind of hypothalamic activator produced by cyclic adenosine monophosphate in pituitary cells [85]. It is generally believed that PACAP could mediate the cyclic adenosine monophosphate (cAMP) signaling pathway through EPAC, a type of cAMP sensor protein, thereby activating the activation of the p38 mitogen-activated protein kinase (MAPK) signaling pathway, stimulating the production of c-Fos, and inducing the expression of *FSHB* [86]. PACAP could also selectively affect the synthesis of FSH through autocrine or paracrine signaling involving follistatin [87]. In addition, it has been reported that GnRH could significantly increase the expression of PACAP and PAC1R in LβT2 cells [88]. Due to the pulsatile release of GnRH, an increasing number of researchers have speculated that PACAP and its receptor, PAC1R, may have different effects in response to different frequencies of GnRH stimulation, which will cause the level of FSH secretion to fluctuate [89].

### 3.6. Transcriptional Regulation of FSH Synthesis and Secretion

#### 3.6.1. Activator Protein-1

Activator protein-1 (AP-1) refers to a dimeric transcription factor formed by c-Jun and c-Fos in the form of a homologous dimer or heterodimer. It has been confirmed that GnRH can induce c-Fos, c-Jun, ATF and other AP-1 transcription factors’ expression in vivo [90] and in vitro in gonadotropin cell lines [91,92] through the MAPK signaling pathway [93] and can then stimulate the transcription of *FSHB* [94]. PACAP-mediated regulation of *FSHB* expression also requires the participation of c-Fos [86]. However, the role of AP-1 transcription factors remains controversial in different species in the process of GnRH regulation of *FSHB* expression. In 2001, Huang et al. [95] provided different opinions on whether AP-1 has the same importance in the regulation of the *FSHB* promoter by GnRH in sheep as in rats and other mammals. The large expression of c-Jun dimer protein can even cause early reproductive senescence in female animals [96]. Therefore, for different species, there may still be some potential differences and unknowns in the regulation of *FSHB* expression and FSH secretion by AP-1 transcription factors.

#### 3.6.2. FOXL2

The *FOXL2* gene, a member of the forkhead transcription factor family, is specifically expressed in adult ovarian granulosa cells [97]. It is related to sex determination and maintenance [98], premature ovarian failure [99], infertility [100], tumors [101,102], and other processes. The *FOXL2* gene is also known as the “denatured gene hidden in the human body” because it exists on non-sex chromosomes, but has the function of maintaining gender characteristics. *FOXL2* plays an important role in regulating FSH secretion, which is essential for the activin-induced transcription of *Fshb* in mice and humans [103]. At present, plenty of studies have confirmed that FOXL2 could coregulate the transcription and expression of *FSHB* by interacting with SMAD3, SMAD4, c-Jun and other proteins after it responds to the signal stimulation of activin [104,105,106]. However, the pathogenic *FOXL2* C134W mutation will change the DNA binding specificity and even drive the generation of adult granulosa cell tumors [107,108]. As an activin-stimulated FSH synthesis regulator, the discovery of *FOXL2* is very important to the field of FSH secretion research. Its effect may be comparable to that of the role of *EGR1* in LH secretion regulated by GnRH, but the question of how *FOXL2* regulates the expression of *FSHB* and the secretion of FSH still needs to be answered by subsequent studies [70,109].

#### 3.6.3. Single Nucleotide Polymorphisms (SNPs) in the FSHR and *FSHB* Genes

In recent years, there have been a number of articles on the role of SNPs in the FSHR genes, some of which have shown that the two very common SNPs at positions 307 and 680 in the 10th exon of the FSHR genes are known to influence the efficiency of signal transduction and are closely related to the ovarian response to in-vitro fertilization (IVF) in the clinic [110,111,112,113]. In addition, studies on granulosa cells have confirmed that the SNPs of FSHR are associated with the transcription activity of the promoter and the binding capacity of FSHR with FSH [114]. Thus, the above-mentioned SNPs of FSHR affect the basic level of FSH in different ways. Grigorova et al. [115] also found a potential regulatory SNP (rs10835638) at 211 bp upstream of the transcription start site of *FSHB* mRNA. This SNP, located in the highly conserved region of placental mammals, has varying degrees of influence on the serum FSH and LH levels, testicular volume, sperm density and many other markers of male reproductive function [115]. Subsequently, it has been proven that the SNP of *FSHB* have a certain effect on serum FSH concentration in cattle [116]. The current research on SNPs of FSHR and *FSHB* genes only reflect the tip of the iceberg, and there might be more SNPs from regulatory factors that regulate FSH secretion. For example, SNPs in *SLC18A2* and *LHX3* might affect transcriptional regulation of *FSHB*, and the SNPs located in *GDNF* or *CXCL12,* which impact FSHR signaling [117,118,119]. An increasing number of studies have provided evidence that SNPs in FSHR can change the secretion levels of FSH and LH, and are closely related to male fertility [120,121,122].

### 3.7. Post-Transcriptional Regulation of FSH Synthesis and Secretion

#### 3.7.1. Non-Coding RNA

Non-coding RNA, a type of RNA that is not capable of encoding the protein, includes certain RNAs with unknown functions and a variety of RNAs with known functions such as rRNA, tRNA, snRNA, snoRNA, microRNA, lncRNA and circRNA [123]. It is widely involved in the processes of FSH synthesis and secretion in mammals as a research focus of post-transcriptional regulation. It is considered that the non-coding RNA, as a key participant and bridge, participates in a variety of different gonadotropin signaling pathways after years of research on post-transcriptional regulation of the *Fshb* gene and related pathways [124] (shown as Figure 1). According to previous studies, many miRNAs, such as miR-21-3p [125], miR-433 [125], miR-186-5p [126], and miR-7a-5p [127] can participate in regulating the secretion of FSH in the manner of inhibiting the expression level of the *Fshb* gene. In addition, lncRNA [128] and circRNA [129] can participate in the regulation of *Fshb* gene expression and FSH secretion through the molecular sponge mechanism of miRNA. For example, lncRNA-m433s1 can reduce the inhibitory effect of miR-433 on *Fshb*, and further regulate FSH secretion by sponging miR-433 [130]. However, there is a large unknown space that researchers need to explore because of the weaker study of non-coding RNA, mRNA stability, RNA methylation modification and other post-transcriptional regulation of FSH secretion mechanisms compared to the transcription level.

#### 3.7.2. Chromatin and Histone Modification

In the past decade, significant progress has been made in understanding how the structure of chromatin changes dynamically to induce or inhibit gene transcription. The chromatin and histone modifications involved in the transcription of *FSHB* have also been gradually discovered. The post-translational modification of histone tails will change the expression of genes to varying degrees through various modifying enzymes (including acetylation, methylation, phosphorylation, glycosylation, etc.) [136]. A large number of chromatin and transcription factor modifications are involved in the process of different GnRH pulse frequencies to stimulate the differential synthesis and secretion of FSH [137,138]. A mass of studies have confirmed that the gonadotropin-specific gene promoters, *Cga*, *Gnrhr*, *Lhb* and *Fshb*, are regulated not only by developmental transcription factors, but also by the epigenetic mechanisms of chromatin structure regulation and histone modifications in αT3-1 and LβT2 cells [139]. Oride et al. [140] found that trichostatin A (TSA), a selective inhibitor of mammalian histone deacetylase, can specifically stimulate the expression of *Fshb,* and confirmed that the two expression mechanisms of gonadotropin subunit genes induced by TSA and GnRH are completely different in LβT2 cells. This also indirectly proves that histone modification plays a potential role in the regulation of FSH synthesis.

Focusing only on the process of GnRH-induced FSH synthesis will find that, in addition to activating specific transcription factors, GnRH can also stimulate chromatin changes through its membrane-bound receptors, thereby promoting the transcription of its target gene *FSHB*, including a variety of epigenetic regulations such as histone and DNA modification changes, nucleosome positioning, and chromatin packaging in gene regulatory regions with the rapid development of epigenetic research [141,142,143]. Even RNA methylation, a kind of epigenetic modification in the spotlight, may also play a role in the process of GnRH regulating FSH synthesis and secretion. In the future, there may be more currently unknown epigenetic modifications that we discover as involved in the regulation of FSH synthesis and secretion.

## 4. GnRH-Regulated FSH Synthesis and Secretion Signaling Pathways

The signaling pathways regulating *FSHB* expression and FSH synthesis are very complex, especially in the process of GnRH regulation of FSH, and multiple signaling cascades are activated simultaneously [39]. In this review, we mainly describe the GnRH regulation of FSH secretion as the entry point and elaborate on the classic signaling pathways activated by GnRH, such as the cAMP/PKA/CREB signaling pathway, PKC/MAPK signaling pathway and Ca^2+^/CaMK II signaling pathway (Figure 2).

### 4.1. cAMP/PKA/CREB Signaling Pathway

The most classic and clearest way for GnRH to regulate FSH secretion is by activating the cAMP/protein kinase A (PKA)/CREB signaling pathway [152]. To date, a large number of studies have explored and verified the molecular mechanism between GnRHR and the cAMP signaling pathway after GnRH stimulation. The generally recognized regulatory mechanism is that, when GnRHR receives GnRH stimulation, it recruits a large amount of Gα and activates the production of cAMP under the action of adenylate cyclase (AC) [153]. The secondary messenger, cAMP, will bind to the regulatory subunits (R subunits) of PKA, which is a kind of tetramer and activate the other two subunits, called the catalytic subunits (C subunits). C subunits will activate CREB to phosphorylate and bind to the CRE site of the *FSHB* regulatory sequence, thereby enhancing the expression level of the *FSHB* gene and promoting the secretion of FSH [144]. To support this molecular regulatory mechanism, several studies have confirmed that GnRH can stimulate the mass production of cAMP and PKA, whether in primary rat pituitary cells or in cell lines such as LβT2 and αT3-1 [154,155].

However, the above-mentioned studies are more focused on exogenous GnRH treatment to explore signaling pathways activation. There are still some differences compared to pulsed release under physiological conditions. Therefore, subsequent studies have gradually shifted from single exogenous GnRH treatment to exploring the dynamic response of rat primary pituitary cells or LβT2 cells after pulsed GnRH treatment. Tsutsumi et al. [145] proved that GnRH pulse stimulation of either a high or low frequency could cause an increase in cAMP and PKA activation. Moreover, CREB will be phosphorylated with each GnRH pulse at a low pulse frequency, but at a higher pulse frequency and amplitude, the phosphorylation of CREB will slow down and become constant [145]. This also shows that the cAMP/PKA/CREB signaling pathway is preferentially activated under low-frequency GnRH stimulation to promote the transcription of *FSHB* and the secretion of FSH. However, the disorder in the expression of any factor in the signaling pathway will lead to changes in downstream physiological activities and even the occurrence of diseases. For example, CREB activation defects can cause spermatogenesis damage, which has been proven in testicular Sertoli cells of rats [156].

### 4.2. PKC/MAPK Signaling Pathway

The MAPK pathway is an important signaling system that mediates cell responses. It is ubiquitous in a variety of organisms and participates in the processes of cell growth, cell development, cell division, cell apoptosis, intercellular functions and other processes [157]. As a class of serine/threonine kinases, MAPK is composed of a variety of isoenzymes. Since extracellular regulated protein kinase (ERK) was identified in 1991, c-Jun N-terminal kinase (JNK)/stress-activated protein kinase (SAPK), p38 and other MAPK subfamilies have also been discovered in mammalian cells [158].

Many studies have shown that GnRH induces the activation of protein kinase C (PKC) in cells, which can activate a variety of MAPK cascades [146], and then MAPKs translocate to the nucleus and activate multiple transcription factors to participate in the regulation of *FSHB* transcription [159]. According to reports, the activation of family members, MAPK1/3 (ERK1/2), MAPK8/9 (JNK1/2) and MAPK14 (p38-α) all mediate the GnRH-induced transcription of *FSHB* [147,160]. These MAPK cascades, especially the ERK signaling pathway, have been recognized to regulate the activity of the *FSHB* promoter in response to GnRH pulse stimulation [161]. In particular, ERK activation occurred more rapidly, activation was more sustained, and the level of nuclear phosphorylated ERK was also higher under stimulation with a low GnRH pulse frequency [161]. However, it was found that the specific knockout of ERK1/2 only partially impaired the expression of *FSHB* and FSH secretion, but the effect on the expression of *Lhb* and LH secretion was more obvious through in vivo verification after the construction of FSH-specific ERK1/2 double knockout mice [162]. This may also imply that the regulatory effect of ERK1/2 may be more important for LH than for FSH, or there may be other GnRH-activated signaling pathways that can compensate for the loss of ERK1/2 function in double knockout mice to maintain the secretion of FSH.

It is worth noting that, since inflammation or oxidative stress can activate the MAPK signaling pathway to a certain extent, the new concept of reactive oxygen species (ROS) as an intermediate signal transduction product of the GnRH-induced response has also been proposed, and it has been confirmed that GnRH stimulation can increase intracellular ROS through NOX/DUOX mediation, thereby enhancing the promoter activity and mRNA level of *FSHB* [163]. This discovery further reveals that ROS generated by other processes may also affect the possibility of GnRH stimulating FSH expression, and enriches the molecular mechanism of the MAPK signaling pathway in the process of GnRH-mediated regulation of FSH secretion. In addition, other studies have shown that the MAPK signal pathway regulates the production of steroids in ovarian granulosa cells and the maturation of oocytes and the changed activity of the MAPK signaling pathway plays a complex regulatory role in testicular damage [164,165,166].

### 4.3. Ca^2+^/CaMK II Signaling Pathway

The rapid increase in intracellular calcium induced by GnRH is also crucial for the expression of *FSHB* and the secretion of FSH, in addition to the two types of classic signaling pathways mentioned above [167]. Ca^2+^ activates a variety of Ca^2+^-dependent signal transduction pathways through a large number of calmodulins to participate in the process of GnRH-mediated regulation of FSH secretion; for example, the combination of GnRH and GnRHR can initiate a variety of cascades involving Ca^2+^ [168]. It has been confirmed that the blockade of Ca^2+^ channels can prevent the increase in *Fshb* expression in rat pituitary cells [169]. Interestingly, the differential regulation of related gene expression is very similar to the effect of pulsed GnRH stimulation in rat primary pituitary cells perfused with Ca^2+^ channel agonists, that is, the low-frequency pulse of GnRH preferentially stimulates the expression of *FSHB*, but under high-frequency stimulation, it preferentially promotes the expression of *Lhb* [170]. The expression level of *FSHB* under high-frequency stimulation does not change as obviously as that in a low-frequency stimulation [171].

Calmodulin-dependent protein kinase II (CaMK II), a common factor that decodes calcium signals and pulse frequencies in many cells, also mediates the process of GnRH regulation of FSH through the Ca^2+^/CaMK II signaling pathway [148]. A single GnRH pulse can induce rapid activation of CaMK II both in rat primary pituitary cells and LβT2 cells [39]. The inhibition of CaMK II will inhibit the expression of *Cga*, *Lhb*, *Fshb* and other gonadotropin related genes [172]. In addition, the activation of MAPK and certain PKC subtypes also requires an increase in the intracellular calcium levels [173]. While the downstream factors of calcium in the regulation of FSH secretion have not been fully elucidated, these research results all prove that CaMK II may play a potential role in decoding the GnRH pulse frequency and regulating the molecular mechanisms of *FSHB*. Therefore, the change of intracellular calcium or CaMK II levels directly lead to the changes of FSH release, and are also associated with the occurrence of several diseases other than reproduction [174,175].

## 5. Conclusions and Prospects

The normal reproductive function and reproductive ability of animals depend on the precise regulation of various reproductive hormones, including FSH. Therefore, it may help us better understand the physiological and pathological processes, such as spermatogenesis, ovulation, the menstrual cycle, puberty, and even reproductive system diseases, if the molecular mechanisms that regulate the synthesis of gonadotropins can be determined. In recent decades, we have clarified the molecular mechanism of the overall regulation of FSH secretion. Every step of FSH synthesis and secretion is strictly controlled by the signals that mediate initial synthesis to the signals required to successfully perform biological functions. Therefore, the regulation of FSH secretion is a highly complex and multilevel network. GnRHR differentially activates a number of different signal transduction pathways in response to changing GnRH pulse frequencies. Various signaling pathways will also interweave and interfere with each other. All of these phenomena further increase the complexity of the molecular mechanism regulating FSH secretion. There are still more kinds of apparent genetic modifications that play unknown functions, although we have clarified the regulation of the FSH molecular basis and signaling pathways. Additionally, more comprehensive studies are needed to decipher the intertwined potential molecular mechanisms that regulate the synthesis and secretion of FSH in different physiological systems. These studies will help us have a clearer understanding of the internal processes regulating animal reproduction, improve the artificial regulatory system of animal reproductive processes, and even provide deeper theoretical support for the exploration and development of potential therapeutic targets and effective therapies related to reproductive diseases or other diseases affected by FSH.

## Figures and Tables

**Figure 1 animals-11-01134-f001:**
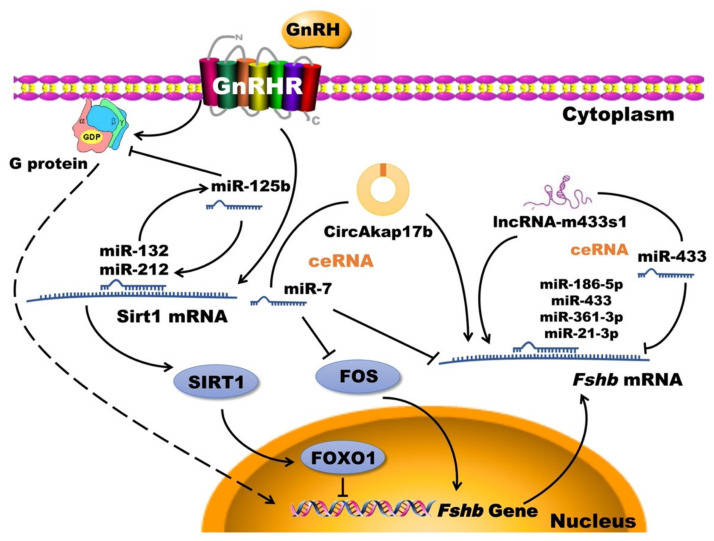
Schematic of known function non-coding RNAs in the post-transcriptional regulation of FSH synthesis. In gonadotroph, GnRH stimulates the synthesis of FSH by directly binding to its receptor GnRHR through several signaling pathways. Some miRNAs (miR-132, miR-212, miR-125b and miR-7) play a role in the gonadotropin pathways [127,131,132,133]. Several miRNAs (miR-186-5p, miR-433, miR-361-3p, miR-7 and miR-21-3p) have been identified to directly target the *Fshb* 3′UTR [125,126,127,134]. Furthermore, circAkap17b [135] and lncRNA-m433s1 [130] up-regulated *Fshb* as miR-7 and miR-433 sponge respectively. ceRNA: competing endogenous RNA; *FOXO1*: forkhead box O1; GnRH: gonadotropin-releasing hormone; GnRHR: gonadotropin-releasing hormone receptor; SIRT1: silent information regulator 1.

**Figure 2 animals-11-01134-f002:**
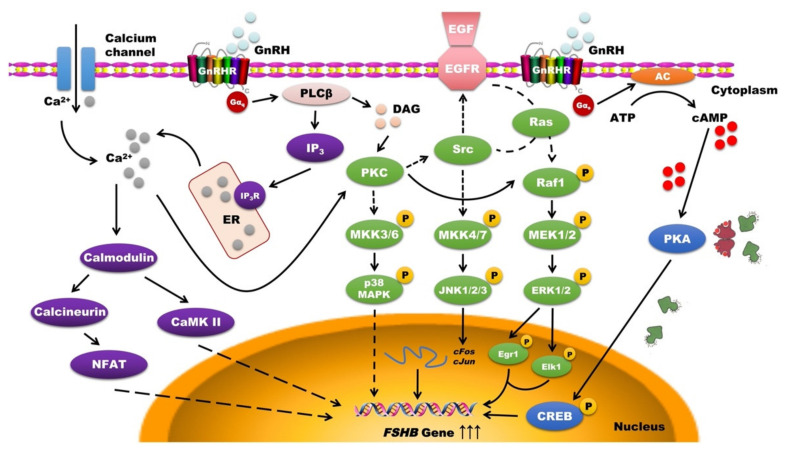
Schematic of classic signaling pathways activated by GnRH. The classic signaling pathways include the cAMP/PKA/CREB signaling pathway, PKC/MAPK signaling pathway and Ca^2+^/CaMK II signaling pathway [144,145,146,147,148,149,150,151]. AC: adenylate cyclase; AP-1: activator protein-1; ATP: adeosine triphosphate; CaMK II: calmodulin-dependent protein kinase II; cAMP: cyclic adenosine monophosphate; CREB: cAMP response element binding; DAG: diacylglycerol; EGF: epidermal growth factor; EGFR: epidermal growth factor receptor; Egr1: early growth response protein 1; ER: endoplasmic reticulum; ERK: extracellular regulated protein kinase; GnRH: gonadotropin-releasing hormone; GnRHR: gonadotropin-releasing hormone receptor; IP_3_: inositol triphosphate; IP_3_R: inositol triphosphate receptor; JNK: c-Jun N-terminal kinase; MAPK: mitogen-activated protein kinase; MEK: MAPK/ERK kinase; MKK: mitogen-activated protein kinase kinase; NFAT: nuclear factor of activated T-cells; PKA: protein kinase A; PKC: protein kinase C; PLC: phospholipase C.

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
