# Peer review of "Advances in the Regulation of Mammalian Follicle-Stimulating Hormone Secretion"

_animals, 2021, doi:10.3390/ani11041134_

Round 1
Reviewer 1 Report
After correcting the manuscript, I have no comments on the current version
Reviewer 2 Report
The manuscript has been vastly improved with the latest revision. The number of primary research articles is increased.
I have only minor concerns
Line 45 "many reproductive disorders"--there is only one that is noted which is PCOS. I feel with "many" more than one should be described.
Line 59. This revised line is simplistic and vague. Revise to something meaningful. activities, "thereby exerting specific biological effects on target tissues[6-9]. " Suggest stating what specific effects and target tissues are being referred to.
Line 67. The phrase " and progesterone in follicular fluid" should be deleted. The sentence is referring to progesterone synthesis, follicular fluid does not synthesize progesterone. If it is important that the readers know that progesterone is measurable in follicular fluid--the sentence needs revision to differentiate synthesis and content.
Line 271 - 275 (and throughout) be consistent in how receptors are referred to. In these few sentences there is "FSH receptor" as well as "FSHR".
Line 277. Suggest "proven" rather than "proved"
Reviewer 3 Report
The Authors fulfilled what was required.
The review is suitable for publication
Reviewer 4 Report
I just have a few suggestions:
L.117 Fshβ refers to a gene in murine models? if not, the name of the gene should be capitalized. In both cases the correct way to write the genes is in italics. The same comment L.132
L.L.248, 251, FOXL2 gene. Gene symbols are italicized, review the complete document regarding this comment.
Author Response
Please see the attachment.

This manuscript is a resubmission of an earlier submission. The following is a list of the peer review reports and author responses from that submission.
Round 1
Reviewer 1 Report
Line 83 says " GnRH is released in a pulsed manner under physiological conditions........", You should briefly explain what physiological factors are involved, what factors stimulate the pulsatile nature of GnRH secretion
Line 96/97 states" ........ or its highly-active analogs". What active analogs, Should be specified.
It would be more appropriate to provide the item number in the references next to the author instead of at the end of the sentence (e.g., line 202, Marina Grigorova et al. [72]; line 123, Szabo et al. [36] )
Reviewer 2 Report
The current version of the submitted review article entitled: “Advances in the Regulation of Mammalian Follicle-Stimulating Hormone Secretion”, is focused to discuss different molecular mechanisms and signaling pathways of FSH synthesis regulation as an important part of mammalian reproductive development. Indeed, through highlighting the nuclear/endocrine cross-talk, different innovative therapeutic approaches could be developed.
The introduction needs to be more developed.
At the part of “structure and function of FSH”, it would be suitable to divide it into 2 parts (first for female reproduction and second for male reproduction), developing each part deeper.
Avoid explaining at the end of some parts of this article, the goal of this review. Put the objectives of your review at the end of the introduction part at once. However, at the end of each part, you should give your personal reflection about citing various data.
Moreover, citing in each part some examples of pathological disorders linked to dysregulation of some molecular signaling pathways (like in hypogonadism for example) and some examples of therapeutic approaches of hormonal stimulations (especially GnRH antagonist) used in animal biotechnology and assisted reproductive technologies, is needed to show the importance of each hormone, gene, and signaling pathway.
Kisspeptin is needed to be added to this article and its role in regulation/stimulation and the recent therapeutic approaches based on it used in male and female infertility problems.
Figure 1 and 2 needed to be dimensioned adequately to the article, explaining the abbreviations at the endnote of the figures
There are some grammatical errors within the entire document that prevent ease of understanding. It needs to be corrected.
Reviewer 3 Report
The manuscript titled Advances in the Regulation of Mammalian Follicle-Stimulating Hormone Secretion by Wang and co-workers is submitted as a review article. The manuscript is well-written, seems thorough, and is interesting. I am troubled the the extensive use of review articles in the the review. I feel review articles are a review of the primary literature and not a review of reviews. It is important to acknowledge the primary literature and to review that literature and not count on previous reviewers to 1) accurately represent the original data and 2) their interpretation of that data may be dated and the primary literature should be reviewed.
Reviewer 4 Report
In this review Wang et al. present a synopsis of all the mechanisms of FSH regulation, highlighting differences between different animal species if any, which is the only feature that justifies its novelty compared to the following review. https://doi.org/10.3389/fendo.2019.00305.
The work may be useful for those who need a general overview that is not too difficult to read. In particular, some parts are not too thorough, such as the paragraph “3.5.3. Single nucleotide polymorphisms (SNPs) in the FSHR and Fshβ genes, that should be expanded given its importance.
Morevoer, some sentence should be rewritten because are not very clear.
Pag 2 line 53: "which may be to regulate the endocrine function of the rat pancreas via the FSH receptor" Please write again this sentence.
Pag 3 line 114: “role in regulating the FSH levels of follicles by” Authors should written more clearly this sentence that is to say: the quantity of FSH that reach the follicle.
Pag 3 line 116: “In male animals, inhibin is closely related to the sperm count[33], sperm concentration[34] and testicular volume[33] secreted by Sertoli cells” .this sentence is not clear, i.e.: in male animals, Inhibin, secreted by Sertoli cells, is closely related to ……
Pag 3: lines 140-143: how would this change in the phosphorylation of cAMP response element binding work? Please explain
Pag 4 line 175: a kind of AP-1 transcription factor[46] should be removed because the Authors have already spoken about c-fos.
pag 5 line 202 : only the surname.
Pag. 5 lines 215-217: It is considered that the miRNA as a key participant and bridge that participates different gonadal signaling pathways after years of research on the translational regulation of FSH.-encoding genes and related pathways[74] (shown as Fig.1). This sentence is not clear.
Fig. 1: the characters are very small and difficult to read
Pag 6 line 246: Aki ORIDE only the surname
Pag 6 line 246-250.Aki Oride et al. found that trichostatin A 246 (TSA) can specifically stimulate the expression of Fshβ, and confirmed that the two expression mechanisms of gonadotropin subunit genes induced by TSA and GnRH are completely different by treating LβT2 cells with TSA, a selective inhibitor of mammalian 249 histone deacetylase[84]. This sentence is not very clear
Pag 7 line 285 maybe a comma between cell in cell.
Pag 8 line 329: in 2015 ????
Pag 8 line 330: stimulating
Pag 8 line 350-351. The inhibition of CaMK II will cause the expression of genes such as Cga, LH. and FSH. to be inhibited[107].This sentence is not clear.
In conclusion, in the present form, the review cannot be published and needs major revisions.
Reviewer 5 Report
The article is very well written, with an adequate structure. Regarding the references, they are actual and relevant.
Round 2
Reviewer 2 Report
The current corrected version of the submitted review article entitled: “Advances in the Regulation of Mammalian Follicle-Stimulating Hormone Secretion”, is focused to discuss different molecular mechanisms and signaling pathways of FSH synthesis regulation as an important part of mammalian reproductive development. Indeed, through highlighting the nuclear/endocrine cross-talk, different innovative therapeutic approaches could be developed.
All the suggested corrections were done. Indeed, the current corrected version of this article is much more improved and enriched with deep scientific reflection offering more interest for scientists in reproduction field.
Reviewer 3 Report
The review manuscript titled Advances in the Regulation of Mammalian Follicle Stimulating Hormone Secretion by Wang and co-workers is a revised manuscript. My primary concern with the original manuscript is that the primary research that contributes to our understanding of FSH secretion was not cited. The authors instead overwhelmingly cited reviews. In the revised manuscript there are many review publications that have been cited, but the citations of the primary literature is improved. There are still concerns with the manuscript. In particular I find most of the revisions to be vague and lacking important detail. Some of the revisions are not germane to the review. The addition of kisspeptin is a review of the regulation of GnRH—not FSH. Only the final paragraph of the kisspeptin review is relevant to the current discussion.
Most egregious are citations that seem incorrect or misinterpreted. For example citation 89 is a review article it is cited in line 224. I search the article for FSH or Follicle Stimulating Hormone—those words were not in the article. Further citations 111 and 112 clearly conflict each other and highlight a controversy of the role of activing in transcription of Fshβ—this is not discussed in the present article. Line 294 states an “increasing number of studies” yet cites only one [128] which is not a review and that publication concludes that a combination of 2 SNPs were associated with FSH and LH, but the other four SNP did not appear to play a significant role in male fertility.
Other considerations
Line 47. The term “significantly” is redundant with “lower”—which implies significance
Lines 56 – 59. Vague and lacks detail
Line 61. It is not clear what “wall cells” refers to. Please clarify.
Line 107 – 108. Vague and lacking detail “Different GnRH pulse frequencies have different effects on the synthesis and release of FSH.” This same vague proclamation is repeated in the next few sentences. This needs detail.
Line 128. What is meant by “is a kind of peptide hormone”? It either is or isn’t a peptide hormone.
Line 196-199. Vague and lacks detail.
Line 285-288. Vague and lacks detail. “varying degrees of influence” needs elaboration and support. Citation 123 is a small clinical trial in men and would not reflect necessarily “male reproductive function”.
There are no citations associated with Figure 1 or 2. Clearly these figures, if not taken from another publication, are reporting other’s data.
Line 349. It is not known what a “hot epigenetic modification” is. Clarify. Also the final sentence (line 350-351) Is vague and poorly written. Certainly there are things that are not known.
Line 355. It is not clear what “This article” is referring to. The current review of the previously cited article. Revise for clarity.
Line 354. Suggest “complex” rather than “complicated”
Line 396. It is not clear what “in downstream life activities” means. Revise for clarity.
Line 462. “several diseases” the citations suggest two.
Reviewer 4 Report
The Authors put a lot of effort into answering questions from all reviewers, overall improving the quality of the work.
I have only some comments, in particular:
Pag. 2 line 52: "it is very meaningful to learn about how to regulate the synthesis and secretion of FSH": How do the Authors think FSH synthesis and secretion could be regulated?
Pag 2 line 73: The Authors should then recall how different preparations of recombinant FSH on the market could have different effects in the treatment of male infertility depending on the diagnosis of the patient, laying the foundations towards a personalized medicine. Cite the literature.
Pag 4 line 183: I’m sorry reaches. It's not important that the Authors use the same sentence as me, the important thing was to change it because it seemed misleading in the previous version.
The work may be published after these minor revisions.